# Grain Structure and Texture Evolution in the Bottom Zone of Dissimilar Friction-Stir-Welded AA2024-T351 and AA7075-T651 Joints

**DOI:** 10.3390/ma17153750

**Published:** 2024-07-29

**Authors:** Haoge Shou, Yaoyao Song, Chenghang Zhang, Pengfei Zhang, Wei Zhao, Xixia Zhu, Peng Shi, Shule Xing

**Affiliations:** 1College of Intelligent Manufacturing, Huanghuai University, Zhumadian 463000, China; 20212249@huanghuai.edu.cn (H.S.); pengshi@huanghuai.edu.cn (P.S.); 2School of Materials Engineering, Jiyuan Vocational and Techncal College, Jiyuan 459000, China; 0001316@jyvtc.edu.cn (W.Z.); zhuxixia@jyvtc.edu.cn (X.Z.); 3Logistics Service Center, Huanghuai University, Zhumadian 463000, China; 20222268@huanghuai.edu.cn; 4Ningbo Institute of Technology, Beihang University, Ningbo 315800, China; 5Xi’an Rare Metal Materials Institute Co., Ltd., Xi’an 710016, China; 6Center for Engineering and Technology, Huanghuai University, Zhumadian 463000, China; 20171670@huanghuai.edu.cn

**Keywords:** friction stir welding, aluminum alloys, dissimilar joint, grain structure, shear texture

## Abstract

High-strength dissimilar aluminum alloys are difficult to connect by fusion welding, while they can be successfully joined by friction stir welding (FSW). However, the asymmetrical deformation and heat input that occur during FSW result in the formation of a heterogeneous microstructure in their welded zone. In this work, the grain structure and texture evolution in the bottom zones of dissimilar FSW AA2024-T351 and AA7075-T651 joints at different welding speeds (feeding speeds) were quantitatively investigated. The results indicated that dynamic recrystallization occurs in the bottom zones of dissimilar FSW joints, and equiaxed grains with low grain sizes are formed at the welding speed of 60–240 mm/min. A high fraction of the recrystallized grains were generated in the bottom zones of the joints at a low welding speed, while a high fraction of the substructured grains are produced at a high welding speed. Different types of shear textures are produced in the bottom zones of the joints; the number fraction of shear texture types depends on different welding speeds. This study helps to understand the mechanism of microstructure homogenization in dissimilar FSW joints and provides a basis for further improving the microstructure of the welded zone for engineering applications.

## 1. Introduction

The use of high-strength aluminum alloys is an effective way to achieve a light weight in aerospace and other fields [1]. Different materials in different parts are usually chosen based on manufacturing processes and costs in practical engineering applications, so joins between dissimilar aluminum alloys are inevitable [2]. Due to the distinctions in physical, chemical, metallurgical, thermodynamic and other performances of AA2024 and AA7075 alloys, it is very hard to join these two alloys by conventional fusion welding methods [3,4,5], which results in the generation of voids, cracks and other fusion welding defects. As a desirable welding method, friction stir welding (FSW) with lower residual stress [6,7] and higher welding efficiency than fusion welding can realize a close connection between these two alloys [8,9], which can be applied to fuselage skin in the aerospace field. Nevertheless, it is easy to produce defects such as holes, tunnels and flash in the process of FSW, which are mainly caused by the mismatch of welding parameters such as rotation speed, welding speed and axial pressure [6]. Therefore, it is necessary to find the matching welding process parameters to prepare the joint without defects. In fact, although defect-free joints can be prepared by optimizing welding process parameters, the intricate thermodynamic coupling characteristics during FSW lead to the formation of a non-homogeneous microstructure in the welding zones of the joints [10,11]; this has a distinct impact on the local microstructure and the bulk mechanical properties of the joints.

The FSW process is comparable to a torsional or extrusion process. The shear textures are prone to being produced in face-centered cubic (FCC) metal after shear deformation [12,13,14,15,16]. Shen et al. [17] concluded that different types of shear textures are formed in the upper and lower areas of the joints due to the occurrence of different degrees of shear deformation. Wang et al. [18] also found that various shear textures are generated in the welding zone of FSW AA6061-T6 joints at different rotation speeds. Imam et al. [19] found that different types of shear textures appear in the top and bottom regions of FSW AA6063 joints and present periodic variations in the onion ring region. Because of the asymmetrical deformation and heat input on both sides of the joints in the process of FSW, different local areas in the joints are subjected to varying thermal deformation behaviors, which eventually produce a heterogeneous microstructure in the same welding areas [20], significantly affecting the overall mechanical properties, especially for dissimilar FSW joints; this microstructural asymmetry can be more conspicuous [21,22]. Therefore, it is imperative to explore the microstructure and texture evolution behavior of the local welding area in dissimilar joints.

Although the grain structure and texture types in the welded regions of joints have been quantitatively analyzed in a previous work [23], the local region involves multiple regions, and the related research on the grain structure and texture components in other regions is still insufficient. Since the bottom region is connected with the substrate and is also the initial location of the fracture reported by our previous work [24], it is of great significance to study microstructure and texture evolution in the bottom region. In this work, a quantitative examination of the grain structure and texture evolution in the bottom region of dissimilar AA2024/7075 joints is performed by electron back-scattering diffraction (EBSD) on the basis of a previous work [25].

## 2. Experimental Details

AA2024-T351 and AA7075-T651 plates of the same size (300 mm long and 40 mm wide) were placed in the advancing side (AS) and retreating side (RS), respectively. The rolling direction (RD) of the two BMs was perpendicular to the welding speed (WD) and parallel to the transverse direction (TD). The tool manufactured from H13 steel possessing a tapered thread pin profile measuring 5 mm pin length and 6 mm shoulder diameter was employed for the welding. During FSW, the tool tilt was set to be 2.5°. Dissimilar FSW butt joints of AA2024-T351 and AA7075-T651 were acquired from 6.5 mm thick plates, which was conducted by using three types of welding speed (feeding speed) of 60, 100 and 240 mm/min at a constant tool rotational speed of 1300 rpm. The cross section of the dissimilar FSW joints perpendicular to the WD was used for microstructure characterization. The EBSD method was performed by employing a field emission scanning electron microscope (FESEM, TESCAN MIRA 3) equipped with an HKL-EBSD system. This EBSD experiment platform with a step size of 0.3 μm was operated at 20 kV, and the Channel 5 software was used for data processing. Prior to EBSD, the joints were ground and polished by a standard metallographic process, and finally electropolished at 15 V and 90 s in a mixed solution of perchloric acid and alcohol in a volume ratio of 1:9. The microhardness measurements were made at the center of the joint on the cross sections perpendicular to the WD. The distance between the two microhardness test points was set to 1 mm.

## 3. Results and Discussion

As shown in Figure 1, the morphologies of the cross section in the welded zone of the three joints present a basin shape, and the two sides in the welded zone of the joints have different degrees of darkness: the dark zone in the left side is AA2024-T351, while the light zone in the right side is AA7075-T651, which results from the fact that AA2024-T35 is more vulnerable to corrosion than AA 2024-T351 [14]. Additionally, AA2024-T351 takes up most of the welded zone, which is attributed to the movement of the stirring tool from the AS to the RS. Based on previous research results, the strip-like coarse grains along the RD are found in the two base materials (BMs), and AA2024-T351 presents a $\left\{001\right\}\langle 100\rangle$ Cube texture, while the $\left\{011\right\}\langle 211\rangle$ Brass and $\left\{123\right\}\langle 634\rangle$ S components are found in AA7075-T651 [10]. As displayed in Figure 2, abundant equiaxed grains are formed in the bottom zone of the three joints at different welding speeds due to the occurrence of dynamic recrystallization resulting from violent plastic deformation and high heat input during FSW.

Figure 3 draws the grain size distribution in the bottom zone of the three joints. The calculated average grain sizes in the bottoms of the three joints at 60, 100 and 240 mm/min are 1.72 ± 0.88 μm, 1.46 ± 0.71 μm and 1.52 ± 0.67 μm, respectively. It can be found that the average grain size in the bottom of the three joints is decreased in turn with increasing welding speed from 60 mm/min to 240 mm/min. This mainly depends on the heat input produced by different welding speeds. During FSW, the generated heat input can be assessed as follows [26]:(1)$$Q=4\pi^{2}\mu P\omega R^{3}/3V$$
where *Q* is the heat input rate unit weld length (kJ/mm), μ is the friction coefficient, *P* is the pressure (kN), *R* is the shoulder radius (mm) and ω and *V* are the rotational speed (r/min) and welding speed (mm/min), respectively. It can be concluded that a high welding speed produces low heat input, leading to a low grain size in the bottom of the joint at 240 mm/min. Furthermore, the average grain size in the bottom zone is lower than that in the center zone, which is reported from our previous work [23]. The bottom zone is only stirred by the end of the stirring probe, inadequate deformation leads to poor flow and the bottom zone is in contact with the substrate, while fast dissipation results in a high cooling rate. As a result, insufficient deformation and low peak temperature lead to the small grains broken in the bottom region not being able to grow, thus forming fine equiaxed grains. Figure 4 exhibits the misorientation distribution; it can be calculated that the percentages of the low angle grain boundaries (LAGBs) at 60, 100 and 240 mm/min are 14%, 14.2% and 14.7%, respectively.

Figure 5 demonstrates the grain structure features in the bottom zones of the three joints; it can be observed that bulk recrystallized grains are formed at the welding speed of 60 mm/min, while plenty of substructured grains are found at the welding speed of 240 mm/min. The calculated deformed, substructured and recrystallized grains are summarized in Figure 6 and Table 1. The percentage of substructured grains in the bottom zone is 92.65%, and few deformed and recrystallized grains are formed in the bottom zone at 240 mm/min. With a decrease in welding speed, the fraction of the substructured grains is decreased, while the fraction of the recrystallized grains is increased. As the welding speed is decreased to 60 mm/min, the maximum fraction of the recrystallized grains reaches 75.29%, while the minimum fraction of the substructured grains is 23.44%. It can be concluded that a low welding speed results in a high recrystallization degree. This may be because a low welding speed produces high heat input, which contributes to further recrystallization by expending more substructured grains. In addition, Figure 6 and Table 1 show the significant difference in the percentage of recrystallized grains. The above Formula (1) shows that the welding heat input is inversely proportional to the welding speed while keeping the other parameters constant. Therefore, the heat input of the joints between 100 and 240 mm/min is significantly lower than that between 60 and 100 mm/min, resulting in the lower recrystallization degree of the welded zone in the joints between 100 and 240 mm/min than that between 60 and 100 mm/min.

Figure 7 shows the distribution maps of the local misorientation angle by calculating the average misorientation value between each pixel and its adjacent eight pixels, illustrating that local orientation gradient information within grains is introduced by deformation. The local misorientation angle is affected by the local lattice curvature, which means the geometrically necessary dislocation density. Therefore, the dislocation density can be expressed as follows [27]:(2)$$\rho =\alpha \theta_{loc}/ndb$$
where α, $\theta_{loc}$, n, d and b are the grain boundary parameters, local misorientation, the defined area size, the step size of EBSD tests and the Burgers vector, respectively. High local misorientation means high dislocation density. As illustrated in Figure 7 and Figure 8, the local misorientation in the bottom zone at 60 mm/min is lower than that at 240 mm/min. Sufficient dynamic recrystallization occurs in the bottom zone, resulting from high heat input at 60 mm/min by expending more dislocations, resulting in lower local misorientation.

During FSW, the rotation and squeezing of the stirring tool lead to severe shear deformation and high heat input, resulting in the formation of a shear texture. For FCC metals, the main shear texture types are as follows [14,15,16]: $\left\{111\right\}\langle \bar{1}\bar{2}2\rangle \;A_{1}^{*}$, $\left\{111\right\}\langle 11\bar{2}\rangle \;A_{2}^{*}$, $\left\{1\bar{1}1\right\}\langle 110\rangle \;A$, $\left\{\bar{1}1\bar{1}\right\}\langle \bar{1}\bar{1}0\rangle \;\bar{A}$, $\left\{1\bar{1}2\right\}\langle \bar{1}1\bar{2}\rangle \;B$, $\left\{\bar{1}1\bar{2}\right\}\langle \bar{1}\bar{1}0\rangle \;\bar{B}$ and $\left\{001\right\}\langle 110\rangle \;C$. Figure 9 exhibits the distribution maps of simple shear texture types in the bottom zones of the three joints and the statistical results are summarized in Figure 10 and Table 2. For different types of shear texture, the components of $\left\{111\right\}\langle 11\bar{2}\rangle \;A_{2}^{*}$ and $\left\{\bar{1}1\bar{2}\right\}\langle \bar{1}\bar{1}0\rangle \;\bar{B}$ are markedly decreased with an increase in welding speed from 60 mm/min to 240 mm/min, while the other shear texture components show almost no significant changes at different welding speeds. In addition, compared to the other two joints, the total number of different types of shear textures at the bottoms of the joints at 60 mm/min is basically the same as that at 100 mm/min, while a high random texture fraction can be found in the bottom zone of the joint at 240 mm/min, which means that the total number of different types of shear texture is decreased at a high welding speed of 240 mm/min. High heat input is generated at a low welding speed, consuming amount of the substructured grains, which forms more textures. A high welding speed produces low heat input, and insufficient recrystallization results in the formation of more random textures.

The microhardness distribution curves with a “W” shape shown in the cross sections of the joints are shown in Figure 11; the microhardness distribution curves on both sides of the welded zone are not completely symmetric, which mainly results from two kinds of substrates on both sides of the joints. The microhardness of the HAZ on the AS of the three joints is lower than that on the RS; this is because high heat input occurs in the HAZ on the AS than that on the RS. It is found that microhardness of the joint at 60 mm/min is low compared to the other two joints. This is due to the fact that high heat input is produced at a low welding speed based on Formula (1), leading to a higher coarsening degree of the microstructure, including grains (Figure 2), and the precipitated strengthened phases of the HAZ on the AS compared that on the RS. A similar phenomenon on the microhardness curves can also be found in dissimilar FSW AA5083 and AA7075 joints [28,29] with different base material states.

## 4. Conclusions


Dynamic recrystallization occurs in the bottom zones of dissimilar FSW joints, resulting in the formation of equiaxed grains. A low grain size with the range of 1.46–1.72 μm is obtained at the welding speed from 60 mm/min to 240 mm/min.A low welding speed results in the formation of a low fraction of LAGBs and a high fraction of the recrystallized grains, while a high welding speed produces a high fraction of LAGBs and substructured grains in the bottom zone.The $\left\{111\right\}\langle 11\bar{2}\rangle \;A_{2}^{*}$ and $\left\{\bar{1}1\bar{2}\right\}\langle \bar{1}\bar{1}0\rangle \;\bar{B}$ components are markedly decreased with an increase in welding speed from 60 mm/min to 240 mm/min, while the other shear texture components show almost no significant changes at different welding speeds. The total number of shear textures in the bottom zone is decreased at a high welding speed compared to other joints.A low welding speed produces high heat input, which results in a low microhardness value in the HAZ on the AS.


## Figures and Tables

**Figure 1 materials-17-03750-f001:**
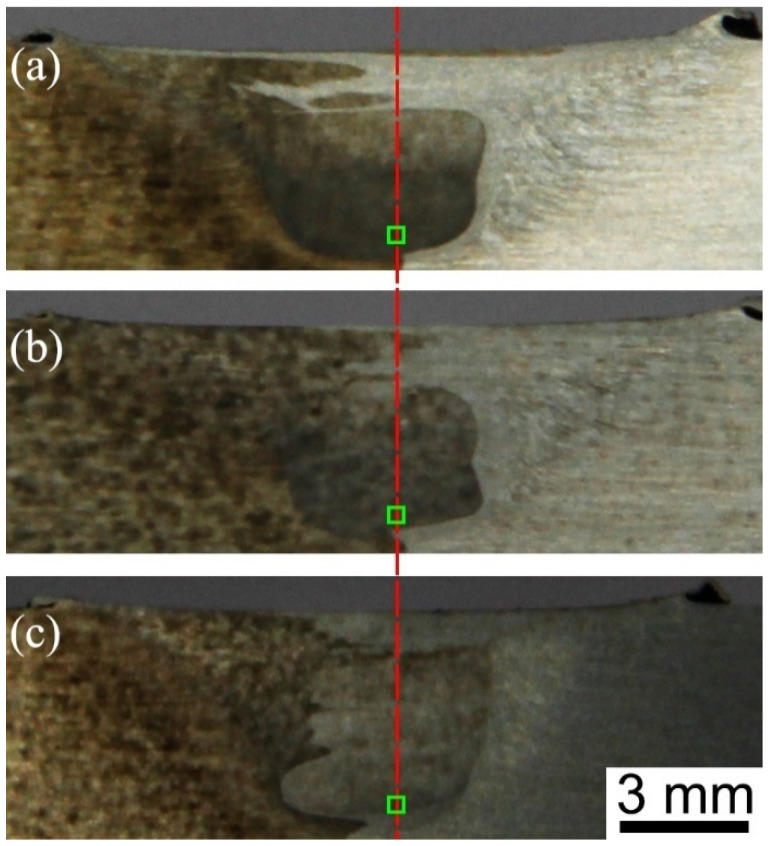
Profiles of cross sections of the three joints: (**a**) 60 mm/min, (**b**) 100 mm/min and (**c**) 240 mm/min [25] (Green square represents the EBSD analysis area).

**Figure 2 materials-17-03750-f002:**
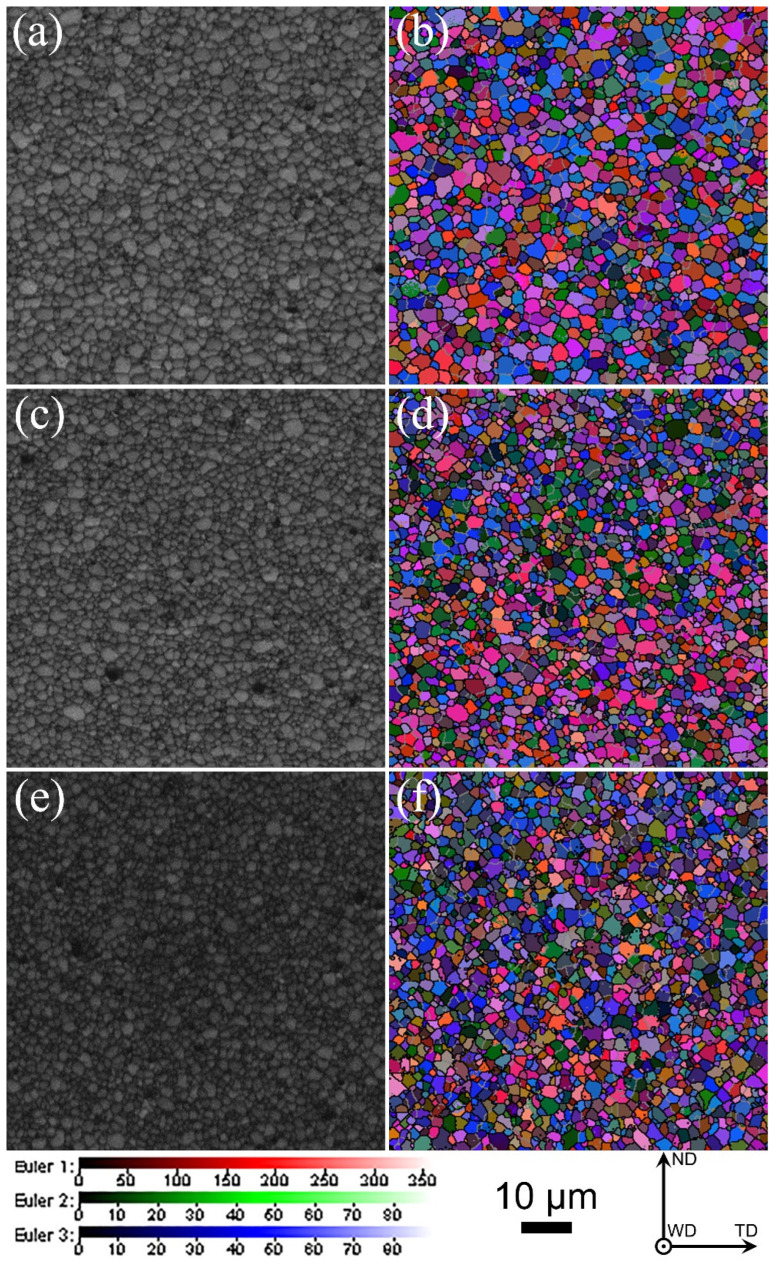
Band contrast and Euler angle (Euler angles 1, 2 and 3 represent the angle/° of rotation around ND, WD and TD, respectively) images of grain structure in the bottom zones of the three joints: (**a**,**b**) 60 mm/min, (**c**,**d**) 100 mm/min and (**e**,**f**) 240 mm/min [25].

**Figure 3 materials-17-03750-f003:**
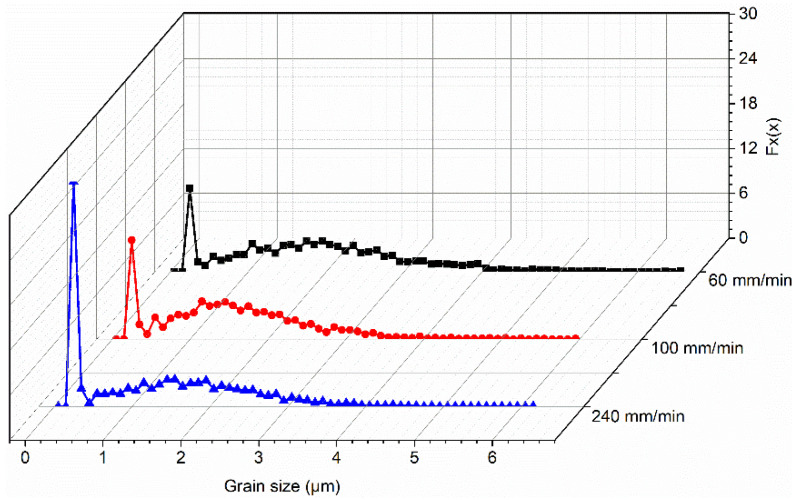
Distribution of grain size in the bottoms of the three joints [25].

**Figure 4 materials-17-03750-f004:**
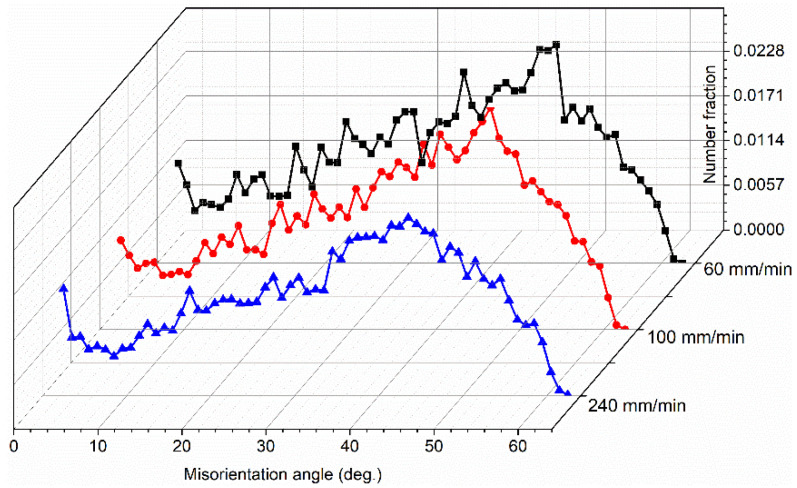
Misorientation angle in the bottoms of the three joints [25].

**Figure 5 materials-17-03750-f005:**
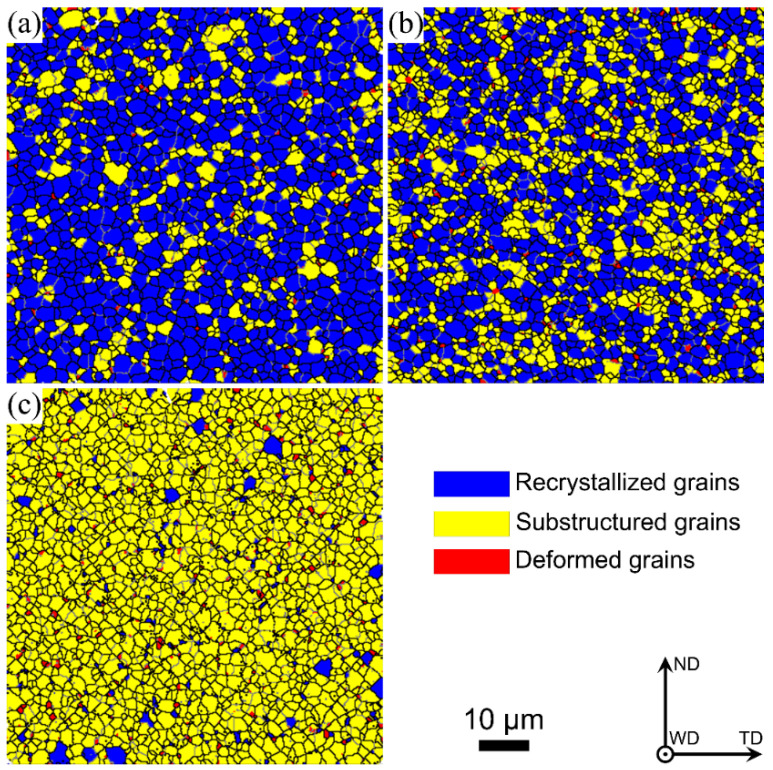
Distribution features of the grains in the bottom zones of the three joints: (**a**) 60 mm/min, (**b**) 100 mm/min and (**c**) 240 mm/min.

**Figure 6 materials-17-03750-f006:**
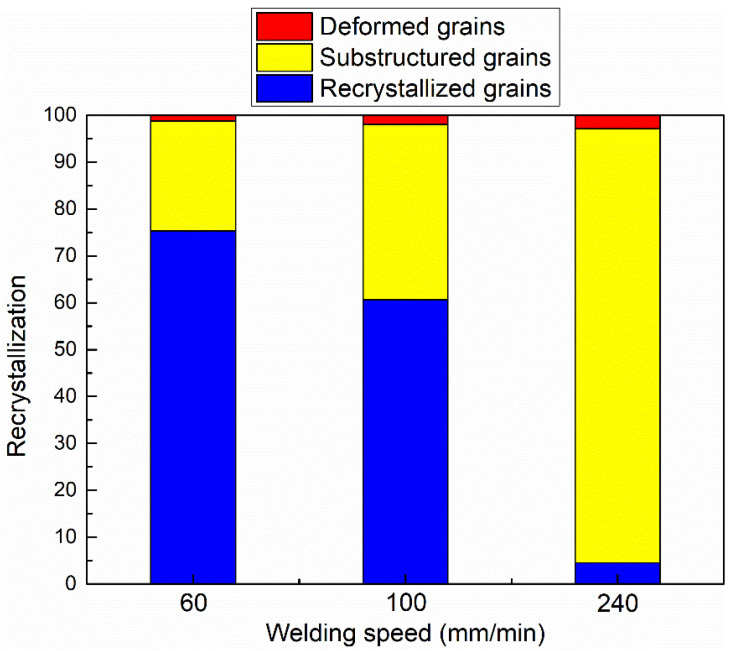
Calculation results of different grain types shown in Figure 5.

**Figure 7 materials-17-03750-f007:**
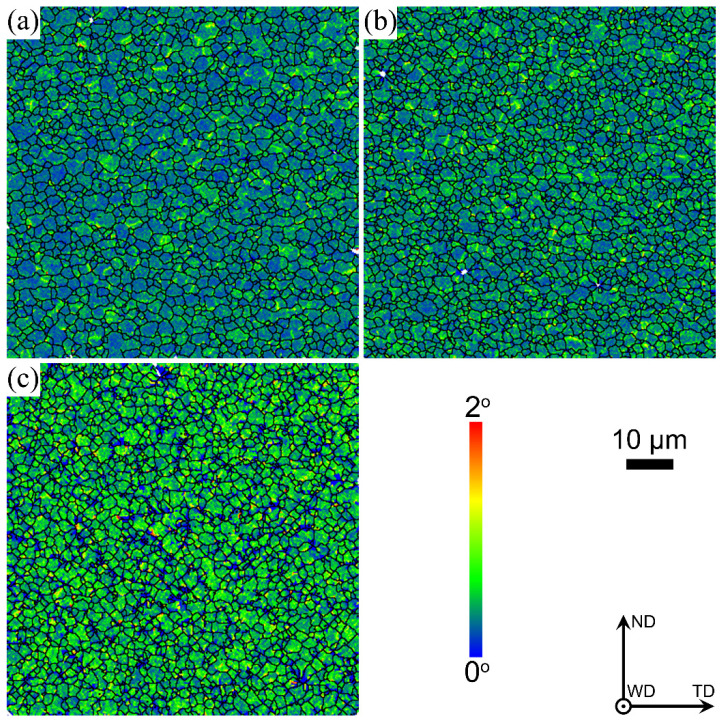
Distribution maps of local misorientation in the bottom zones of the three joints: (**a**) 60 mm/min, (**b**) 100 mm/min and (**c**) 240 mm/min.

**Figure 8 materials-17-03750-f008:**
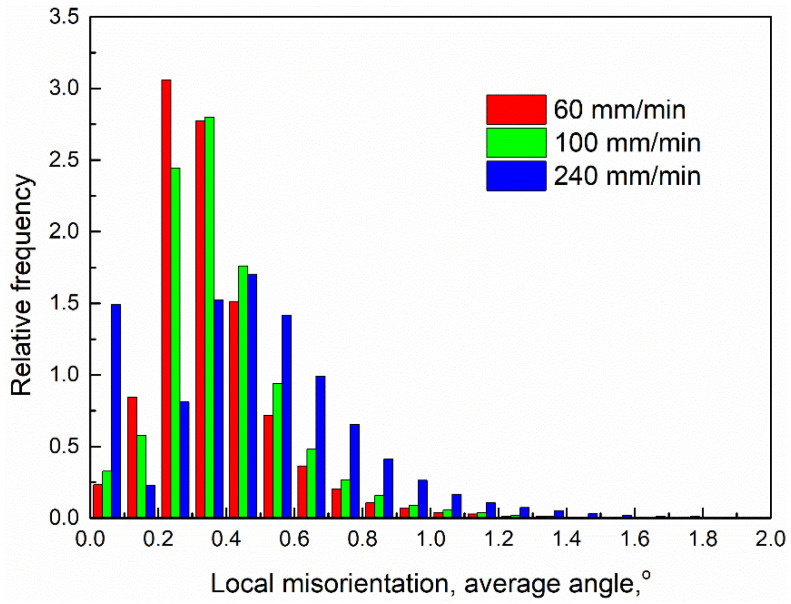
Calculation results of local misorientation shown in Figure 7.

**Figure 9 materials-17-03750-f009:**
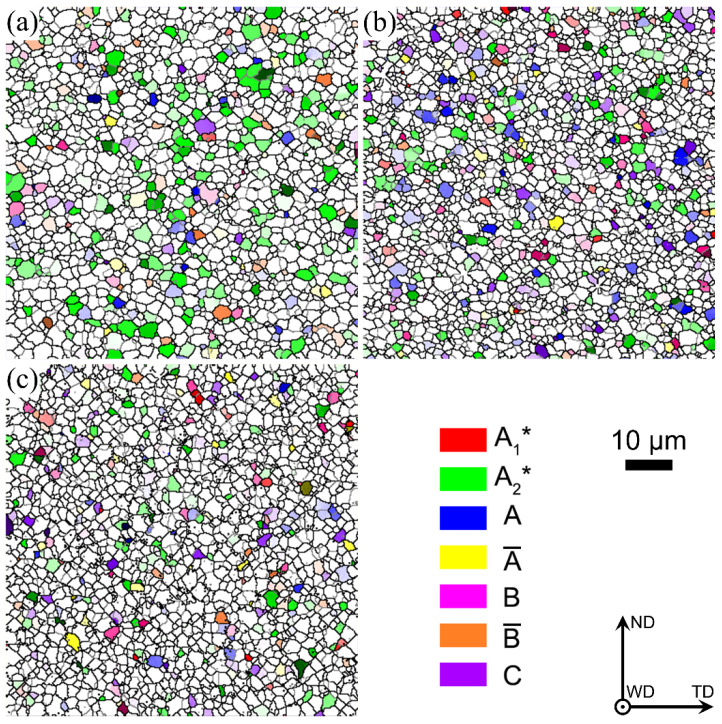
Distribution of shear texture components in the bottom zone of the three joints: (**a**) 60 mm/min, (**b**) 100 mm/min and (**c**) 240 mm/min.

**Figure 10 materials-17-03750-f010:**
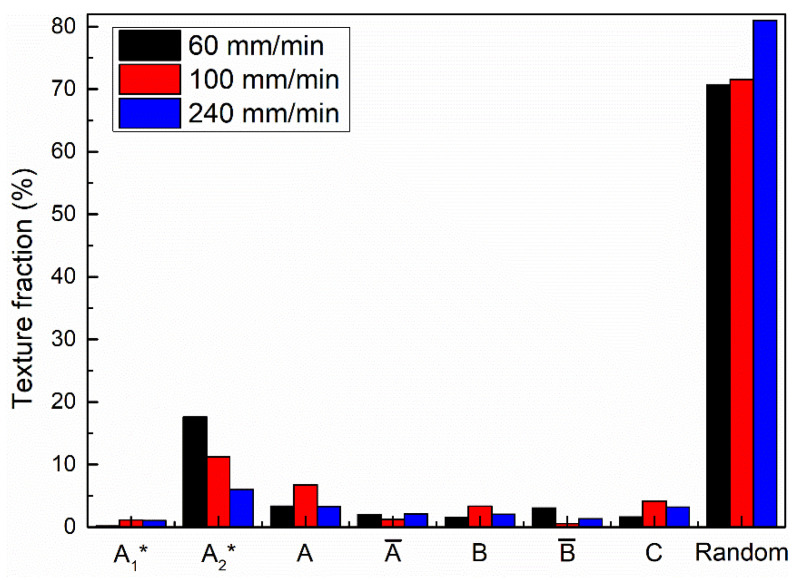
Calculation results of different shear texture components in Figure 9.

**Figure 11 materials-17-03750-f011:**
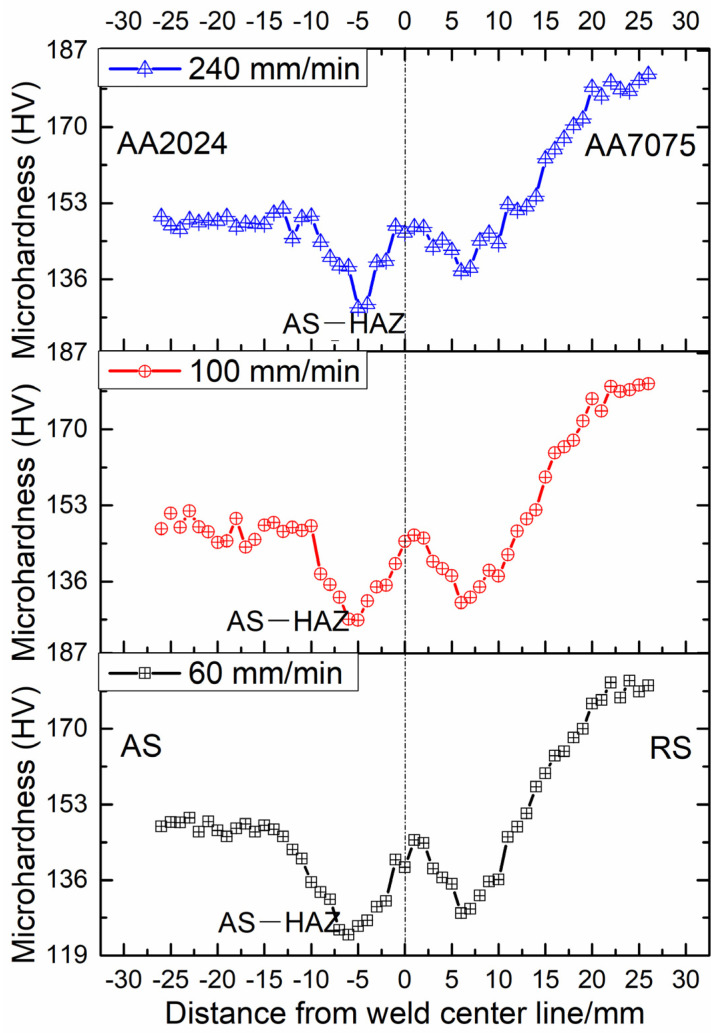
Microhardness distribution of the cross sections of the joints [25].

**Table 1 materials-17-03750-t001:** Statistical analysis results of recrystallized, substructured and deformed grains in Figure 5.

Welding Speed/mm/min	Recrystallized Grains/%	Substructured Grains/%	Deformed Grains/%
60	75.29	23.44	1.27
100	60.59	37.46	1.95
240	4.5	92.65	2.85

**Table 2 materials-17-03750-t002:** Statistical analysis results of different types of shear textures in Figure 9.

Welding Speed/mm/min	Different Types of Shear Texture Components/%							
	$A_{1}^{*}$	$A_{2}^{*}$	A	$\bar{A}$	B	$\bar{B}$	C	Random
60	0.168	17.6	3.35	2	1.51	3.08	1.63	70.662
100	1.16	11.2	6.75	1.25	3.35	0.583	4.17	71.537
240	1.02	5.98	3.29	2.1	2.03	1.34	3.21	81.03

## Data Availability

The original contributions presented in the study are included in the article, further inquiries can be directed to the corresponding author/s.

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
