# Peer review of "Grain Structure and Texture Evolution in the Bottom Zone of Dissimilar Friction-Stir-Welded AA2024-T351 and AA7075-T651 Joints"

_materials, 2024, doi:10.3390/ma17153750_

Round 1
Reviewer 1 Report
Comments and Suggestions for Authors
The introduction provides a solid background on the subject and cites relevant references. It sets the context for the study by discussing the importance of FSW in joining dissimilar aluminum alloys and highlights the existing knowledge gaps that this research aims to address.
The research design is appropriate. The study effectively uses various welding speeds to investigate their impact on microstructural properties and texture evolution in dissimilar aluminum alloy joints.
While the methods are generally well described, including specific details such as the exact setup of the FSW process, parameters like tool rotation speed, tilt angle, and exact welding conditions would enhance reproducibility. Additionally, the process for microstructural and texture analysis, including EBSD settings, could be elaborated.
The results are clearly presented with appropriate use of figures and tables to illustrate the findings. The discussion is well-integrated with the results, providing a coherent narrative of the study's outcomes.
The conclusions are well-supported by the results, summarizing the key findings and their implications effectively. They reflect the data presented and tie back to the study's aims.
Author Response
Comments1:
The introduction provides a solid background on the subject and cites relevant references. It sets the context for the study by discussing the importance of FSW in joining dissimilar aluminum alloys and highlights the existing knowledge gaps that this research aims to address.
The research design is appropriate. The study effectively uses various welding speeds to investigate their impact on microstructural properties and texture evolution in dissimilar aluminum alloy joints.
While the methods are generally well described, including specific details such as the exact setup of the FSW process, parameters like tool rotation speed, tilt angle, and exact welding conditions would enhance reproducibility. Additionally, the process for microstructural and texture analysis, including EBSD settings, could be elaborated.
The results are clearly presented with appropriate use of figures and tables to illustrate the findings. The discussion is well-integrated with the results, providing a coherent narrative of the study's outcomes.
The conclusions are well-supported by the results, summarizing the key findings and their implications effectively. They reflect the data presented and tie back to the study's aims.
Response1:
Thank you very much for your recognition, appreciation and support of our work, and we will continue to work hard for further research.
Reviewer 2 Report
Comments and Suggestions for Authors
The paper is interesting and concerns a practical problem - FSW of aluminum alloys AA2024-T351 and AA7075-T651.
I have the following remarks:
1. I believe that, in addition to a study of the microstructure, an analysis of the mechanical properties of the joints should also be carried out, which has not been done. it is important to relate the material microstructure to the mechanical properties;
2. Fig.5 shows the significant difference in the percentage of recrystallized grains. Further research is needed with a speed between 100 and 240 mm/min.
3. The study does not provide detailed information on the statistical analysis of the data.
4. Many pictures and equations from other studies are used. If they are removed, how many pages will the authors' research be? In this aspect, I believe that the article should be expanded with more diverse studies, considering more scenarios and factors. The article is concise and insufficient to be published in a reputable journal.
5. The research has not produced any practical recommendations for improving the welds of these two materials.
It is my strong belief that the article requires a significant revision to address the outlined issues and enhance its quality.
Author Response
Comments 1:I believe that, in addition to a study of the microstructure, an analysis of the mechanical properties of the joints should also be carried out, which has not been done. it is important to relate the material microstructure to the mechanical properties;
Response 1:Thank you very much for your comments. The results and the corresponding analysis of the mechanical properties such as microhardness curves of these three joints has been added in this work. The aim of this work is to elucidate the evolution of the grain structure of the local welding area at the bottom of the joints with the process parameters, so as to further explain the relationship between microstructure and mechanical properties of the joints.
Comments 2:Fig.5 shows the significant difference in the percentage of recrystallized grains. Further research is needed with a speed between 100 and 240 mm/min.
Response 2:Thank you very much for your comments. It can be observed from Fig. 5 that significant difference in the percentage of recrystallized grains. The heat input formula is introduced for qualitative interpretation by discussing the significant difference and supplementing the related analysis contents by marking red fonts in the revised manuscript.
Comments 3:The study does not provide detailed information on the statistical analysis of the data.
Response 3: Thank you very much for your comments. The detailed information on the statistical analysis of the data such as recrystallization grains (Table 1) and shear texture (Table 2) components has been supplemented by marking red fonts in the revised manuscript.
Comments 4: Many pictures and equations from other studies are used. If they are removed, how many pages will the authors' research be? In this aspect, I believe that the article should be expanded with more diverse studies, considering more scenarios and factors. The article is concise and insufficient to be published in a reputable journal.
Response 4:Thank you very much for your advice. After our careful consideration, your opinion is very helpful to the promotion of this work. Actually, although some pictures come from other studies used, the presented data and the content to be considered and analyzed are diverse, as you mentioned. Thus, we try our best to supplement the relevant contents including some experimental data and mechanical properties by marking red fonts in the revised manuscript.
Comments 5:The research has not produced any practical recommendations for improving the welds of these two materials.
Response 5:Thank you very much for your advice. Your comments are very pertinent indeed. In fact, a lot of work to improve the performance of the welds of these two materials has been done in our previous work (doi: 10.1007/s11665-018-3785-9, 10.1016/j.msea.2019.138368 and 10.1016/j.jmapro.2019.11.031 et al.) and ideal performance of the welds has been also obtained. The focus of this work is to clarify the evolution law of grain structure in different welding processes, so as to reveal the evolution law of mechanical properties of the welds with process parameters.
Reviewer 3 Report
Comments and Suggestions for Authors
In the Introduction, it is necessary to refer to the defects in the microstructure of this type of welded joint and the factors influencing them. It is necessary to refer to the microhardness profile in the cross-section of the welded joint. Figure 2 - the legend would be better if it were horizontal and the font was larger. In the figure caption, explain the meaning of Euler1 - Euler3, and provide the unit. Figure 3 - it is better to divide each of the graphs into separate component graphs, otherwise they will be invisible, and a larger font is needed. Line 108 - enter the units of the quantities used in the formula (1). Figure 5 - move the legend to the right or down outside the chart - it unnecessarily covers part of the chart area. Figures 7 and 9 - the background of the chart would be better to be uniform (white) Figure 9 - The bar graph would be clearer It is a pity that no microhardness analysis was carried out in the cross-sections of the obtained welded joints It would be worth comparing the obtained results with similar results published in the literature for this or a similar welded joint. It is worth updating the literature (although it is quite new, this research field is developing rapidly).
Comments on the Quality of English LanguageIn the Introduction, it is necessary to refer to the defects in the microstructure of this type of welded joint and the factors influencing them. It is necessary to refer to the microhardness profile in the cross-section of the welded joint. Figure 2 - the legend would be better if it were horizontal and the font was larger. In the figure caption, explain the meaning of Euler1 - Euler3, and provide the unit. Figure 3 - it is better to divide each of the graphs into separate component graphs, otherwise they will be invisible, and a larger font is needed. Line 108 - enter the units of the quantities used in the formula (1). Figure 5 - move the legend to the right or down outside the chart - it unnecessarily covers part of the chart area. Figures 7 and 9 - the background of the chart would be better to be uniform (white) Figure 9 - The bar graph would be clearer It is a pity that no microhardness analysis was carried out in the cross-sections of the obtained welded joints It would be worth comparing the obtained results with similar results published in the literature for this or a similar welded joint. It is worth updating the literature (although it is quite new, this research field is developing rapidly).
Author Response
Comments 1:In the Introduction, it is necessary to refer to the defects in the microstructure of this type of welded joint and the factors influencing them. It is necessary to refer to the microhardness profile in the cross-section of the welded joint. Figure 2 - the legend would be better if it were horizontal and the font was larger. In the figure caption, explain the meaning of Euler1 - Euler3, and provide the unit. Figure 3 - it is better to divide each of the graphs into separate component graphs, otherwise they will be invisible, and a larger font is needed. Line 108 - enter the units of the quantities used in the formula (1). Figure 5 - move the legend to the right or down outside the chart - it unnecessarily covers part of the chart area. Figures 7 and 9 - the background of the chart would be better to be uniform (white) Figure 9 - The bar graph would be clearer It is a pity that no microhardness analysis was carried out in the cross-sections of the obtained welded joints It would be worth comparing the obtained results with similar results published in the literature for this or a similar welded joint. It is worth updating the literature (although it is quite new, this research field is developing rapidly).
Response 1:
Thank you very much for your good advice. These minor comments raised above have been corrected as follows:
1--The defects in the microstructure of the FSW joint and the factors influencing them have been supplemented in the section of Introduction by marking blue fonts.
2--Figure 2 - the legend has been adjusted to the horizontal level, the font has been raised, and the Euler Angle meaning and unit have been added to the figure caption.
3--Figure 3 - the graphs have been divided into separate component graphs and the fonts have been enlarged.
4-- Line 108 - the units of the quantities have been entered in the formula (1).
5--the legend in Fig. 5 has been moved to the top outside the chart.
6--Figures 7 and 9 - the background of the chart has been adjusted to be white.
7--Figure 9 - The graph has been adjusted to be bar graph.
8--The microhardness analysis has been added by marking blue fonts.
9--The obtained results with similar results published in the literature for the similar welded joints have been compared by citing newly published similar literature on dissimilar FSW aluminum alloy joints.
Reviewer 4 Report
Comments and Suggestions for Authors
This paper investigates the microstructure of the bottom zone of the two dissimilar types of AL alloys manufactured by friction stir welding. There are several minor and major issues that should be addressed before it can be accepted from my side.
1. Line 14-16 in the Abstract, should be rewritten. Why is it necessary to joint AA2024 to the AA7075? Please describe it if there is any specific application.
2. In line 18, characterise the welding speeds. Specify whether the welding speeds are rotational speeds or feed rates.
3. Rewrite this sentence, “The results indicated that dynamic recrystallization occurs in the bottom zone of the dissimilar FSW joints, and low grain size of the formed equiaxed grains is formed at the welding speed of 60-240 μm.”
4. In line 25,” in dissimilar FSW joints” is right.
5. In line 30, please describe the applications of these alloys and explain why we need to join them. Additionally, justify why FSW is a better joining method compared to others, such as adhesive joints and fusion welding. For example The residual stresses in FSW welds are lower than those generated during fusion welding [*]
[*] Farhang, M., Farahani, M., Nazari, M., & Sam-Daliri, O. (2022). Experimental Correlation Between Microstructure, Residual Stresses and Mechanical Properties of Friction Stir Welded 2024-T6 Aluminum Alloys. International Journal of Advanced Design & Manufacturing Technology, 15(3).
[**]Butola, R., Choudhary, N., Kumar, R., Mouria, P. K., Zubair, M., & Singari, R. M. (2021). Measurement of residual stress on H13 tool steel during machining for fabrication of FSW/FSP tool pins. Materials Today: Proceedings, 43, 256-262.
6. Why the bottom zone microstructure in the welding area is important. This is something that you investigated in your paper. This should be indicated in the introduction.
7. In Fig. 2, it appears that increasing the welding speed from 60 mm/min to 240 mm/min reduces the average grain size. How do the welding speeds affect the defects in the weld and bottom zones? How does the hardness change due to various welding speed?
8. In the experimental section (line 74), the authors mentioned that an FESEM was employed, but there are no SEM images included in the results. By including SEM images of the cross-sectional area, you can observe any defects or damage resulting from different welding speeds in the manufacturing process.
9. The conclusion is too short. You need to briefly summarize the important results of the paper.
Author Response
Comments 1:
Line 14-16 in the Abstract, should be rewritten. Why is it necessary to joint AA2024 to the AA7075? Please describe it if there is any specific application.
Response 1:Thank you very much for your comments and questions. Line 14-16 in the Abstract has been rewritten and the AA2024/7075 dissimilar joints can be applied for the fuselage skin in the aerospace field, both of them have been added in the revised manuscript.
Comments 2:In line 18, characterise the welding speeds. Specify whether the welding speeds are rotational speeds or feed rates.
Response 2: Thank you very much for your reminders. The welding speeding refers to feeding speed (it can also be distinguished by the units at the back), which has been corrected in the Abstract and Experiment section in the revised manuscript.
Comments 3:Rewrite this sentence, “The results indicated that dynamic recrystallization occurs in the bottom zone of the dissimilar FSW joints, and low grain size of the formed equiaxed grains is formed at the welding speed of 60-240 μm.”
Response 3:Thank you very much for your comments. The sentence you mentioned above has been corrected in the revised manuscript.
Comments 4:In line 25,” in dissimilar FSW joints” is right.
Response 4:Thank you very much for your comments. “in dissimilar FSW joint” has been presented in the revised manuscript.
Comments 5:In line 30, please describe the applications of these alloys and explain why we need to join them. Additionally, justify why FSW is a better joining method compared to others, such as adhesive joints and fusion welding. For example The residual stresses in FSW welds are lower than those generated during fusion welding [*]
[*] Farhang, M., Farahani, M., Nazari, M., & Sam-Daliri, O. (2022). Experimental Correlation Between Microstructure, Residual Stresses and Mechanical Properties of Friction Stir Welded 2024-T6 Aluminum Alloys. International Journal of Advanced Design & Manufacturing Technology, 15(3).
[**]Butola, R., Choudhary, N., Kumar, R., Mouria, P. K., Zubair, M., & Singari, R. M. (2021). Measurement of residual stress on H13 tool steel during machining for fabrication of FSW/FSP tool pins. Materials Today: Proceedings, 43, 256-262.
Response 5:Thank you very much for your comments. The applications of these alloys and the reason we need to join them In line 30 have been described and explained, respectively. The relative contents and literature you mentioned have been corrected and cited in the revised manuscript.
Comments 6:Why the bottom zone microstructure in the welding area is important. This is something that you investigated in your paper. This should be indicated in the introduction.
Response 6:Thank you very much for your comments. The contents you mentioned above have been supplemented in the last part of the Introduction in the revised manuscript.
Comments 7:In Fig. 2, it appears that increasing the welding speed from 60 mm/min to 240 mm/min reduces the average grain size. How do the welding speeds affect the defects in the weld and bottom zones? How does the hardness change due to various welding speed?
Response 7:Thank you very much for your questions. The welding speed is usually combined with the rotating speed to prepare a defect-free joint, which is a common regulatory parameter in the actual welding process. The regulation of the two does not match, and welding defects such as holes and pores are easy to form in the weld area of the joint. If the welding speed is too fast, some defects such as tunnels and weak connections are easy to occur in the bottom zones. The welding speed with the range from 60 mm/min to 240 mm/min selected in this work is the optimized parameters in our previous work, so almost no defects are found in the bottom zone.
Low welding speed produce high heat input, leading to high coarsening degree of microstructure including grains and the precipitated strengthened phases, which results in low microhardness value (Fig. 11 in the revised manuscript).
Comments 8:In the experimental section (line 74), the authors mentioned that an FESEM was employed, but there are no SEM images included in the results. By including SEM images of the cross-sectional area, you can observe any defects or damage resulting from different welding speeds in the manufacturing process.
Response 8: Thank you very much for your questions. The FESEM mentioned was employed is mainly used in conjunction with EBSD system for the analysis of grain structure, which has been elaborated in the Experiment section. Since the welding speed with the range from 60 mm/min to 240 mm/min selected in this work is the optimized parameters in our previous work, almost no defects are found in the bottom zone. Thus, SEM images is not required to observe defects in this work.
Comments 9:The conclusion is too short. You need to briefly summarize the important results of the paper.
Response 9:Thank you very much for your questions. The conclusion has been briefly summarized in the revised manuscript.
Round 2
Reviewer 2 Report
Comments and Suggestions for Authors
My remarks have been taken into account by the authors in the revised version. I recommend publishing of the paper as is.
Reviewer 4 Report
Comments and Suggestions for Authors
Accept in present form